# ADAPTIVE AUTOMOTIVE RADAR DATA ACQUISITION

## ABSTRACT

In an autonomous driving scenario, it is vital to acquire and efficiently process data from various sensors to obtain a complete and robust perspective of the surroundings. Many studies have shown the importance of having radar data in addition to images since radar improves object detection performance. We develop a novel algorithm motivated by the hypothesis that with a limited sampling budget, allocating more sampling budget to areas with the object as opposed to a uniform sampling budget ultimately improves relevant object detection and classification. In order to identify the areas with objects, we develop an algorithm to process the object detection results from the Faster R-CNN object detection algorithm and the previous radar frame and use these as prior information to adaptively allocate more bits to areas in the scene that may contain relevant objects. We use previous radar frame information to mitigate the potential information loss of an object missed by the image or the object detection network. Also, in our algorithm, the error of missing relevant information in the current frame due to the limited budget sampling of the previous radar frame did not propagate across frames. We also develop an end-to-end transformer-based 2D object detection network using the NuScenes radar and image data. Finally, we compare the performance of our algorithm against that of standard CS and adaptive CS using radar on the Oxford Radar RobotCar dataset.

## 1 INTRODUCTION

The intervention of deep learning and computer vision techniques for autonomous driving scenario is aiding in the development of robust and safe autonomous driving systems. Similar to humans navigating their world with numerous sensors and information, the autonomous driving systems need to process different sensor information efficiently to obtain the complete perspective of the environment to safely maneuver. Numerous studies Meyer & Kuschk (2019), Chang et al. (2020) have shown the importance of having radar data in addition to images for improved object detection performance. The real-time radar data acquisition using compressed sensing is a well-studied field where, even with sub-Nyquist sampling rates, the original data can be reconstructed accurately. During the onboard signal acquisition and processing, compressed sensing will reduce the required measurements, therefore, gaining speed and power savings. In adaptive block-based compressed sensing, based on prior information with a limited sampling budget, radar blocks with objects would be allocated more sampling resources while maintaining the overall sampling budget. This method would further enhance the quality of reconstructed data by focusing on the important regions. In our work, we split the radar into 8 azimuth blocks and used the 2D object detection results from images as prior data to choose the important regions. The 2D object detection network generates the bounding boxes and object classes for objects in the image. The bounding boxes were used to identify the azimuth of the object in radar coordinates. This helped in determining the important azimuth blocks. As a second step, we used both previous radar information and the 2-D object detection network to determine the important regions and dynamically allocate the sampling budget. The use of previous radar data in addition to object information from images mitigates the loss of object information either by image or the object detection network.

Finally, we have also developed an end-to-end transformer-based 2-D object detection Carion et al. (2020) network using the NuScenes Caesar et al. (2020) radar and image dataset. The object detection performance of the model using both Image and Radar data performed better than the object detection model trained only on the image data.

Our main contributions are listed below:

- We have developed an algorithm in a limited sampling budget environment to dynamically allocate more sampling budget to important radar return regions based on the prior image data processed by the Faster R-CNN object detection network Ren et al. (2015) while maintaining the overall sampling budget.

- In the extended algorithm, important radar return regions are selected using both object detection output and the previous radar frame.

- We've designed an end-to-end transformer-based 2-D object detection network (DETR-Radar) Carion et al. (2020) using both Nuscenes Caesar et al. (2020) radar and image data.

## 2 RELATED WORK

The compressed sensing technique is a well-studied method for sub-Nyquist signal acquisition. In Roos et al. (2018), they showed the successful application of standard compressed sensing on radar data with 40% sampling rate. They also showed that they could use a single A/D converter for multiple antennas. In our method, we have shown efficient reconstruction with 10% sampling rate. In another work, Slavik et al. (2016), they used standard compressed sensing-based signal acquisition for noise radar with 30% sampling rate. Whereas, we've developed this algorithm for FMCW scanning radar. Correas-Serrano & González-Huici (2018) analyzed various compressed sensing reconstruction algorithms such as Orthogonal Matching Pursuit (OMP), Basis Pursuit De-noising (BPDN) and showed that, in automotive settings, OMP performs better reconstruction than the other algorithms. In our algorithm, we have used Basis Pursuit(BP) algorithm for signal reconstruction because, although it is computationally expensive than OMP, BP requires fewer measurements for reconstruction Sahoo & Makur (2015).

The Adaptive compressed sensing technique was used for radar acquisition by Assem et al. (2016). They used the previously received pulse interval as prior information for the present interval to determine the important regions of the pulsed radar. In our first algorithm, we used only the image data and in the second algorithm, we used both image and previous radar data on the FMCW scanning radar. In another work, Kyriakides (2011) they used adaptive compressed sensing for a static tracker case and have shown improved target tracking performance. However, in our algorithm, we've used adaptive compressed sensing for radar acquisition from an autonomous vehicle where, both the vehicle and the objects were moving. In Zhang et al. (2012) they used adaptive compressed sensing by optimizing the measurement matrix as a separate least squares problem, in which only the targets are moving. This increases the computational complexity of the overall algorithm. Whereas in our method, the measurement matrix size is increased to accommodate more sampling budget which has the same complexity as the original CS measurement matrix generation technique. In a separate but related work, Nguyen et al. (2019) LiDAR data was acquired based on Region-of-Interest derived from image segmentation results of images. In our work, we acquired radar data using 2-D object detection results.

Apart from radar, adaptive CS was used for images and videos. In Mehmood et al. (2017), the spatial entropy of the image helped in determining the important regions. The important regions were then allocated more sampling budget than the rest which improved reconstruction quality. In another work, Zhu et al. (2014), the important blocks were determined based on the variance of each block, the entropy of each block and the number of significant Discrete Cosine Transform (DCT) coefficients in the transform domain. In Wells & Chatterjee (2018), object tracking across frames of a video was preformed using Adaptive CS. The background and foreground segmentation helped in allocating higher sampling density to the foreground and very low sampling density to the background region. Liu et al. (2011) similarly used adaptive CS for video acquisition based on inter-frame correlation. Also, Ding et al. (2015), performed the joint CS of two adjacent frames in a video based on the correlation between the frames.

Finally, numerous studies showed the advantage of using both radar and images in an object detection network for improved object detection performance. Nabati & Qi (2019) used radar points to generate region proposals for the Fast R-CNN object detection network which made the model faster than the selective search based region proposal algorithm in Fast R-CNN. Nobis et al. (2019), Chadwick et al. (2019) showed that radar in addition to image improved distant vehicle object detec-

tion and occluded object detection due to adverse weather conditions. In Chang et al. (2020), spatial attention was used to combine radar and image features for improved object detection using FCOS Tian et al. (2019) object detection network.

## 3 METHOD

### 3.1 DATASET

The Oxford Radar RobotCar dataset Barnes et al. (2019) consists of various sensor readings while the vehicle was driven around Oxford, the UK in January 2019. The vehicle was driven for a total of 280km in an urban driving scenario. We have used camera information from the center captured with Point Grey Bumblebee XB3 at 16 Hz, rear data captured by Point Grey Grasshopper2 at 17 Hz. The Radar data was collected by NavTech CT350-X, Frequency Modulated Continuous-Wave scanning radar with 4.38 cm range resolution and 0.9 degrees in rotation resolution at 4 Hz. They captured radar data with a total range of 163m. In addition to this, they have released the ground-truth radar odometry. However, in our case, we require 2D object annotation on the radar data to validate our approach. Since this a labor-intensive task, we chose 3 random scenes from the Oxford dataset, each with 10 to 11 radar frames and marked the presence or absence of the target object across reconstructions. To the best of our knowledge, this is the only publicly available raw radar dataset. Hence, we used this dataset for testing our compressed sensing algorithm.

In order to train our DETR-Radar object detection model, we used the NuScenes v0.1 dataset Caesar et al. (2020). This is one of the publicly available datasets for autonomous driving with a range of sensors such as Cameras, LiDAR and radar with 3D bounding boxes. Similar to Nabati & Qi (2019), we have converted all the 3D bounding box annotations to 2D bounding boxes and merged similar classes to 6 total classes, Car, Truck, Motorcycle, Person, Bus and Bicycle. This dataset consists of around 45k images and we have split the dataset into 85% training and 15% validation.

### 3.2 ADAPTIVE BLOCK-BASED COMPRESSED SENSING

Compressed Sensing is a technique where, if the original signal $x$, with $x \in R^N$, the measurement $y$ is taken using the measurement matrix $\phi$, using $y = \phi x$, where, $\phi \in R^{(MXN)}$ and $y \in R^M$ with an effective number of measurements, $M << N$. The theory, Mehmood et al. (2017) guarantees that if the input is K-sparse in some domain, the data, with a high probability can be recovered from $O(KlogN)$ measurements. In order to recover the signal from $y$, we used Basis Pursuit(BP) algorithm. We used binary sparse matrix as the measurement matrix He et al. (2010) since it promotes easy hardware implementation. In our case, we assumed that the data is sparse in DCT domain. The BP algorithm uses $l_1$ norm to recover the data using, $min_x ||\theta x||_1 \ s.t. \ \phi x = y$, where, $\theta$ is the domain transformation matrix. The radar data is split into 50x100 blocks and CS is applied to each block. However, depending on the importance of a block, the number of measurements M is increased dynamically while maintaining the total budget as 10% sampling rate.

### 3.3 ALGORITHM 1

The front camera generates images at 16 Hz, the rear camera at 17 Hz and the radar frames are acquired at 4 Hz. The object detection network takes 0.12s per image to generate bounding boxes and object class for a particular image. This algorithm is designed to access object information from images 0.18s before the radar frame acquisition timestamp to account for processing the image to get the bounding boxes and object classes. The front camera has 66 degrees horizontal Field-of-View (HFoV). The rear camera has 180 degrees HFoV. The radar data covers the entire 360 degrees HFoV. Therefore, at a given timestamp, from the camera image, we have information about the rear surrounding of 180 degrees plus the front of 66 degrees. From this technique, there would be a blind spot of 57 degrees to the left and 57 degrees to the right. However, this would be mitigated in Algorithm 2.

In our method, we are performing an adaptive block-based CS. The radar data is split into 8 equal regions, in azimuth and 37 equal regions in range. This creates a block of size 50x100. Therefore, from the camera image, the important azimuth sections to focus on would be derived based on the presence of objects in that section. Since the depth information is not available from the camera

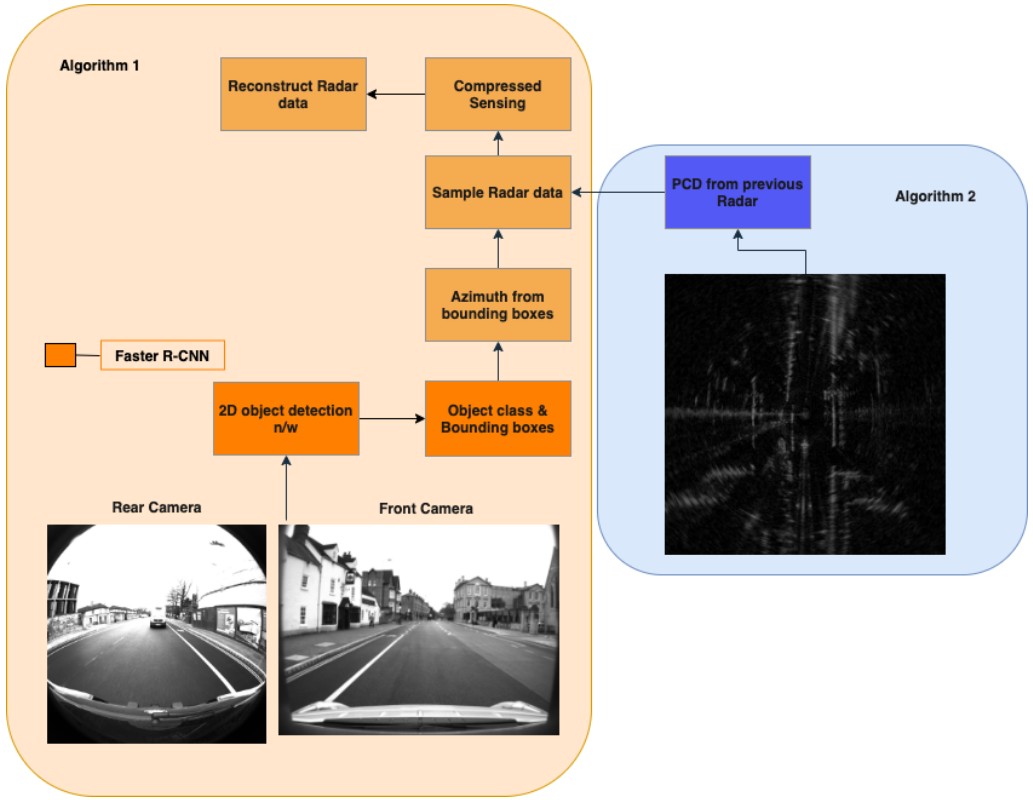

Figure 1: The overview of Algorithm 1 and Algorithm 2. Algorithm 1 takes front and rear camera data as input to identify the important azimuth blocks. Algorithm 2 samples radar data based on azimuth from images and the previous radar frame.

images, in this case, we can only choose azimuth sections and not the range. The chosen sections were dynamically allocated more sampling rate than the others. Also, since the radar data was acquired with 163 m range, we allocated more sampling rate to the first 18 range blocks compared to the last 19 since there is not much useful information in the farther range values. The average driving speed in an urban environment is 40 miles per hour. Since radar is captured at 4 Hz, for every frame, the object could have moved 4.25m. Since the bin resolution is 4.38cm and for a particular block with 100 bins, the area spanned would be 4.38m. Since we are looking into the first 18 range blocks, that covers a total area of 78.84m. Hence, in an urban setting, the moving vehicle can be comfortably captured by focusing within the 78.84m range. In a freeway case, for an average speed of 65 miles per hour, the vehicle could have moved 7.2m per frame and again, this would be captured by focusing on the first 78.84m. Algorithm 1 steps:

1. Acquire front and rear camera information at t=0s.
2. Process them simultaneously through the Faster R-CNN network and generate results at t=0.12s.
3. Process the Faster R-CNN results to determine the important azimuth blocks for radar.
4. Dynamically allocate more sampling rate to the chosen azimuth blocks and fewer to the other blocks while maintaining the total sampling budget.
5. Acquire the radar data at t=0.25s based on the recommenced sampling rate.

### 3.4    ALGORITHM 2

However, in order to avoid the effects of the blind spot region of the camera, the lost information from the camera due to weather or other effects and object missed by the object detection network,

we designed the extended algorithm to utilize the previous radar information to sample the current radar frame, in addition, to object detection results from images. In order to maintain the bit budget, the chosen range blocks for a particular azimuth is further limited to 14, covering 61.32m in range. Based on the speed of the vehicle explanation given in the algorithm 1 section for urban and freeway environment, the moving vehicles can still be captured within the 61.32m range from the autonomous vehicle. Therefore, the balance sampling budget is used to provide a higher sampling rate to the chosen blocks from algorithm 2. Algorithm 2 steps:

1. Repeat steps 1 to 5 from algorithm 1 for frame 1.
2. Repeat steps 1 to 3 from algorithm 1.
3. Using the Constant False Alarm Rate (CFAR) Richards (2005) detection algorithm, the blocks with point clouds from radar data are additionally chosen to have a higher sampling rate.
4. Acquire the radar data based on the above allocation.

As shown in figure 1, Algorithm 1 takes rear and front camera images and predicts object class and bounding boxes. The object's x coordinate in the front camera image is converted from the cartesian plane to azimuth ranging from -33 to 33 degrees corresponding to the HFoV of the front camera. The object's x coordinate from the rear camera is converted to azimuth ranging from 90 to 270 degrees. Therefore, if an object is present in one of the 8 azimuth blocks, divided as 0-45 degrees, 45-90 degrees and so on, that azimuth block is considered important and allocated more sampling rate. Algorithm 2 is an extension of Algorithm 1, in addition to azimuth, if a radar point cloud is detected in any of the blocks (50x100), they are marked necessary and were allocated more sampling rate. In the experiments section, we have shown cases where the radar point clouds helped in identifying objects present in the camera's blind spot.

### 3.5 DETR-RADAR

The Faster R-CNN Ren et al. (2015) object detection network is one of the well-refined techniques for 2D object detection. However, it relies heavily on two components, non-maximum suppression and anchor generation. The end-to-end transformer-based 2D object detection introduced in Carion et al. (2020), eliminates the need for these components. We included radar data in two ways. In the first case, we included the radar data as an additional channel to the image data Nobis et al. (2019). In the second case, we rendered the radar data on the image. We used perspective transformation based on Nabati & Qi (2019) to transform radar data points from the vehicle coordinates system to camera-view coordinates. The models where radar data were included as an additional channel were trained for a longer duration since the first layer of the backbone structure had to be changed to accommodate the additional channel. In all the above cases, the models were pre-trained on the COCO dataset and we fine-tuned them on the NuScenes data. To the best of our knowledge, we are the first ones to implement end-to-end transformer-based 2D object detection using both image and radar data.

## 4 EXPERIMENTS

In order the validate our approach, we tested it on 3 scenes from the Oxford radar robot car dataset. The image data were processed using the Faster R-CNN object detection network Girshick et al. (2018) to obtain object classes and 2D bounding boxes. ResNet-101 was used as the backbone structure, which was pre-trained on the ImageNet dataset Russakovsky et al. (2015). The faster R-CNN network was trained on the COCO train dataset and gave 39.8 box AP on the COCO validation dataset. Since the model was originally trained on the COCO dataset, it predicts 80 classes. However, we filtered pedestrians, bicycle, car and truck classes from the predictions for a given scene. In order to capture and present the radar data clearly, we have shown data for a range of 62.625m from the autonomous vehicle. Although the original radar data is captured for a range of 163m, we could find meaningful information in the first 62.625m. The total sampling budget was set to 10%. Therefore, baseline 1 reconstruction was uniformly provided with 10% sampling rate for the entire frame and were reconstructed using standard CS. In baseline 2, similar to Assem et al. (2016), the important blocks were determined using the pointcloud information from the previous radar frame.

However, the first frame for a scene was acquired with uniform sampling of 10%. From the second frame, if an object was detected using CFAR in the previous frame in a given block, it was deemed essential and the sampling budget to that block was increased. We set the overall budget to 10% and all the non-important blocks had 5% sampling rate. Therefore, the remaining sampling rate was equally distributed to all the important blocks as suggested by the CFAR algorithm.

In algorithm 1, we split the radar into three regions. R1: The chosen azimuth until 18[th] range block (78.84m), R2: the other azimuth regions until the 18[th] range block (78.84m) and finally, R3: all the azimuth blocks from 19[th] to 37[th] range block. Since the object detection on image data does not provide the depth information, for a particular chosen azimuth block, we will be forced to choose the entire range of 163m. However, in the urban and freeway scenario, as explained in the methods sections, the moving vehicle can be comfortably captured within the first 18 blocks (78.84m) and hence we chose the first 18 blocks as the R1 region. The purpose of splitting the regions into R1, R2 and R3 is to provide with decreasing sampling rate based on decreasing importance. Moreover, the regions could have been split as 16 blocks as in the case of Algorithm 2 or 20 blocks as well which would have only changed the sampling rate allocation to different regions. Since across different scenes, there could have been a variable number of important azimuth blocks, if the chosen azimuth block was less than 50% i.e., less than 4 out of 8, we randomly sampled more azimuth blocks. Also, if it was more than 4, we avoided losing important information by reducing the sampling rate in the R1 region. We fixed the sampling rate for R2 and R3 initially. R3 had the lowest sampling rate of 2.5% because it covered the region around the autonomous vehicle that was 78.84m away. Similarly, R2 was closer to the vehicle but based on the image information, it did not have an object and hence we fixed the sampling rate to 5%. Therefore, based on the number of important azimuth blocks, we solved for the sampling rate of R1. Hence, when there were 4 azimuth blocks, R1 was sampled at 30.8%, R2 at 5% and R3 at 2.5% sampling rate. In the case of 5 azimuth blocks, R1 was sampled at 25.5%, R2 and R3 at 5% and 2.5% respectively. In the case of 6 azimuths, R1 at 20.2%, R2 at 5% and R3 at 2.5%.

In algorithm 2, again, the regions were again split into 3. However, R1: The chosen azimuth until 14[th] range (61.32m), R2 was the other azimuth region until 14[th] and R3 were from 15[th] to 37[th] range block. In this case, we retained the sampling rate across regions from algorithm 1. But, we reduced the number of blocks in the R1 region from 18 to 14. Again, although we reduced the R1 region to the first 61.32m, the moving vehicle can be comfortably captured based on the explanation from methods. The previous reconstructed radar frame was processed by the CFAR detector to identify point clouds corresponding to objects. If a radar block had a point cloud, it was deemed important. The sampling budget saved from R1 was used to sample radar blocks classified as important from the previous radar frame. In table 1, we have discussed the presence or absence/faint visibility of an object in a frame across reconstructions. Apart from that, in all frames across scenes except for frame 2, scene 3 truck reconstruction, the mentioned objects were sharper in our reconstruction compared to the baseline 1. The radar images have been enhanced for better visibility of the objects. Due to the non-availability of object annotation from the Oxford radar dataset, we had to annotate the presence or absence of the object manually and have provided a subjective evaluation.

In the first scene, the autonomous vehicle moved across an almost straight street without turning. There was a vehicle behind it and pedestrians on the right sidewalk and a pedestrian on the left sidewalk to the front. As shown in the table 1, the person on the top left is not visible or very faint on both the baseline reconstructions on frame 2 to 6. In frame 7 to 10, the person on the top left was reconstructed by baseline 2 while it was very faint in baseline 1 reconstruction. However, the person is visible on both the Algorithm 1 and Algorithm 2 reconstructions. In frame 2, the pedestrians on the right side happened to be in the blind spot of both the front and rear cameras. Therefore, the reconstruction of Algorithm 1 on that segment was poor. However, that region was captured from the previous radar frame and a higher sampling rate was allocated to the region with pedestrians and it was reconstructed appropriately using Algorithm 2. Apart from the listed objects in the table, the rear vehicle was, in general, sharper than baseline reconstruction in all the frames reconstructed using Algorithm 1 and Algorithm 2.

In scene 2, the vehicle waited on the side of an intersection for a truck to pass by. This truck is visible in all the reconstructions. In frame 2, the car was in the blind spot and it was missed by the CFAR in the previous radar frame as well as the image. Therefore, it is visible in the baseline 1 and it was not reconstructed by our algorithm. In frames 4-6 and 8, the car to the rear right was missed by the camera. But, it was captured in algorithm 2 and was sharper than the baseline 1

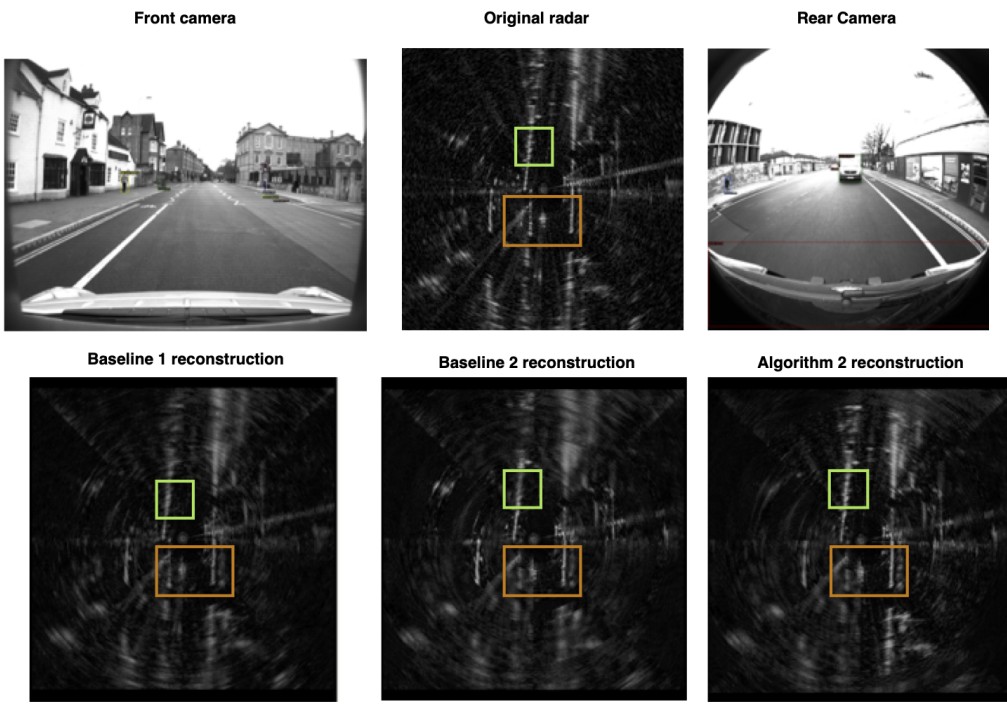

Figure 2: The figure above is from Scene 1, frame 2. In the top row, we show the front image data and rear image processed by the 2D object detection network. The original radar corresponds to the raw radar data acquired in the scene. The green box on the radar data, when zoomed in, would show the person to the left on the front camera. The person is not visible on the baseline reconstructions but can be seen in our reconstruction. The orange box highlights the truck on the rear image.

| Scene | Frame | Object | Baseline1 | Baseline2 | Algo1 | Algo2 |
|---|---|---|---|---|---|---|
| Scene 1 | 1 | Person (top-left) | no | no | yes | - |
| | 2-6 | Person (top-left) | no | no | yes | yes |
| | 7-10 | Person (top-left) | no | yes | yes | yes |
| | 2 | Pedestrians(right) | yes | yes | no | yes |
| Scene 2 | 7 | car (top-left) | no | no | yes | yes |
| | 6-11 | Bicycle (rear) | no | no | yes | yes |
| | 2 | car (rear-right) | yes | no | no | no |
| | 4-6,8 | car (rear-right) | yes | yes | no | yes |
| | 4-7 | Pedestrian (rear-left) | no | no | yes | yes |
| | 9-11 | Car(rear-left) | yes | no | no | yes |
| Scene 3 | 2-6 | car (rear) | no | no | yes | yes |
| | 2,4 | car (top-right) | no | yes | yes | yes |
| | 3 | car (top-right) | no | no | yes | yes |
| | 8 | car (rear-right) | no | no | yes | yes |

Table 1: The table highlights the presence of an object as 'yes' and if the object is very faint or absent, it is indicated as 'no'.

reconstruction. Again, in frames 9-11, the car to the rear left was missed by the object detection network. But, it was captured from the previous radar frame and it was reconstructed by Algorithm 2 with a higher sampling rate. Apart from this the car to the front of the vehicle, pedestrian at the rear of the vehicle were reconstructed by Algorithm 1 and Algorithm 2 while it was either missed or had a faint reconstruction by both the baseline techniques.

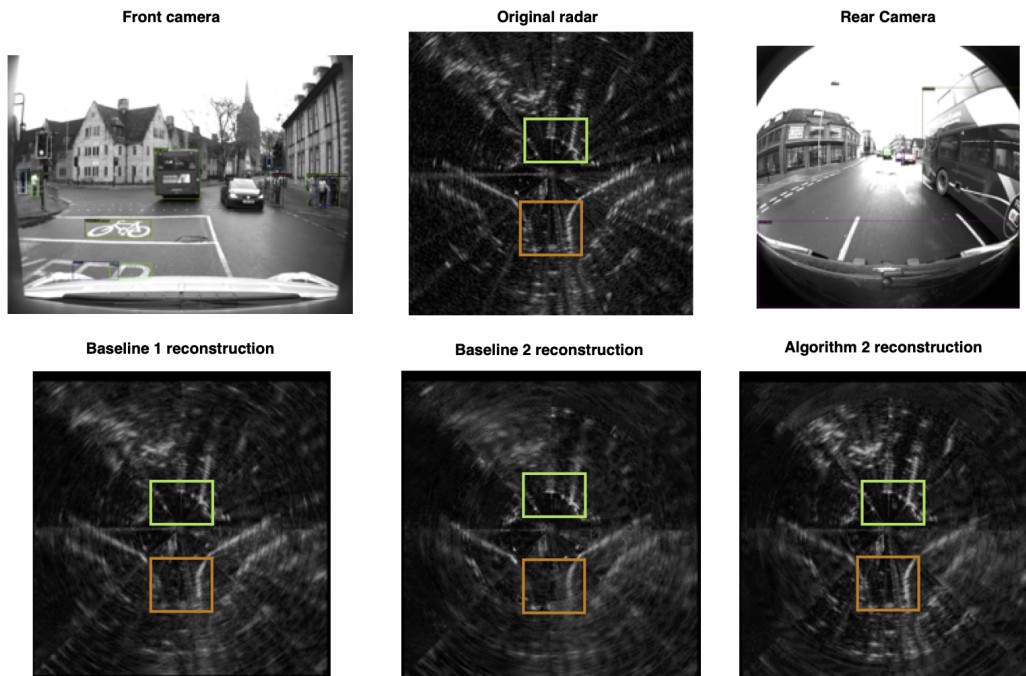

Figure 3: The figure above is from Scene 3, frame 3. The green box when zoomed in shows the car passing by to the right of the autonomous vehicle right below the front bus and radar returns from the wall. The orange box highlights the car behind the autonomous vehicle, captured by our algorithms and missed by the baselines.

In scene 3, the vehicle crossed a traffic signal. There was a bus to the left, another bus to the front and a car passed by on the opposite direction. There were a few cars behind the vehicle. In all the frames, the buses were captured by all the reconstruction schemes. However, our algorithm gave sharper results than baseline 1. As shown in the table 1, the cars behind the autonomous vehicle were captured by our algorithms. However, it was not captured by the baseline reconstructions. The result is highlighted using an orange box in figure 3. Similarly, the car passing by on the opposite side was captured by our algorithm and it is barely visible in the baseline reconstruction and it is highlighted using a green box in figure 3. In raw radar, the poles and buildings tend to have a sharper appearance than cars or pedestrians based on the size or position of the cars and pedestrians. Therefore, the car is the tiny region right below the bus and building, highlighted by the green box. This validates the necessity to allocate a higher sampling budget for important objects such as pedestrians or cars on the road. Moreover, although we used previous radar information in addition to object detection results for radar reconstruction, our algorithm did not exhibit propagation of error of missing an object in the current frame due to poor reconstruction in the previous frame.

Finally, we trained a separate object detection network using the NuScenes image and radar data. In this case, we limited our analysis to NuScenes image and original radar data because the Nuscenes radar that was available to us were processed pointclouds with annotations. Whereas, the Oxford data was the only available raw data on which we could apply CS but, without object annotations and hence, we could not use Oxford data to train the object detection model. All of our models were trained on the COCO detection dataset and we fine-tuned them on the Nuscenes v0.1 dataset. As shown in the table 2, our baseline comparison is with the Nabati & Qi (2019) paper, where, they trained the model on Nuscenes v0.1 image dataset and used the radar data for anchor generation. The Faster R-CNN Img and Faster R-CNN RonImg models had ResNet-101 He et al. (2015) as the backbone structure Girshick et al. (2018). The models with Img+R were trained with radar as an additional channel. Therefore, the first layer of the backbone structure was changed to process the additional radar channel. The DETR network Carion et al. (2020) had ResNet-50 He et al.

(2015) as the backbone structure, a transformer encoder, transformer decoder followed by a 3-layer feed-forward bounding box predictor. The main advantage of transformers is that its architectural complexity is simpler than the Faster R-CNN and the need for Non-Maximal suppression is eliminated in DETR. Also, we believe that the attention mechanism in DETR helped in focusing more on the regions with radar points (overlapping on objects in images) and helped in better performance.

The Img and RonImg models were trained for the same number of epochs for a fair comparison. The Img+R models were trained for additional epochs since the backbone structure's first layer was modified. In the Faster R-CNN case, Img+R has better performance than Img. While, in DETR, RonImg has better performance. The Faster R-CNN Img and RonImg were trained for 25k iterations. The Faster R-CNN Img+R was trained for 125k iterations. DETR Img and DETR RonImg models were trained for 160 epochs. While DETR Img+R was trained for 166 epochs. The DETR - RonImg model performed better across various metrics compared to the baseline, Faster R-CNN and DETR Img+R model. We believe that the attention heads in the transformer architecture helped in focusing object detection predictions around the radar points. However, the Faster R-CNN Img+R was better than the Faster R-CNN RonImg model. We used the standard evaluation metrics, mean average precision (AP), mean average recall (AR), average precision at 0.5, 0.75 IOU, small, medium and large AR Lin et al. (2015).

| Network | AP | AP50 | AP75 | AR | ARs | ARm | ARl |
|---|---|---|---|---|---|---|---|
| Fast R-CNN Nabati & Qi (2019) | 0.355 | 0.590 | 0.370 | 0.421 | 0.211 | 0.391 | 0.514 |
| Faster R-CNN - Img | 0.395 | 0.678 | 0.417 | 0.470 | 0.256 | 0.444 | 0.568 |
| Faster R-CNN - Img+R | 0.462 | 0.738 | 0.503 | 0.530 | 0.328 | 0.515 | 0.599 |
| Faster R-CNN - RonImg | 0.380 | 0.654 | 0.400 | 0.449 | 0.176 | 0.421 | 0.563 |
| DETR - Img | 0.471 | 0.802 | 0.504 | 0.616 | 0.384 | 0.572 | 0.725 |
| DETR - RonImg | 0.486 | 0.804 | **0.527** | **0.636** | 0.401 | **0.602** | 0.731 |
| DETR - Img+R | 0.448 | 0.763 | 0.468 | 0.582 | 0.297 | 0.549 | 0.688 |

Table 2: Img denotes model trained on Images, Img + R indicated model trained with Radar as an additional channel and RonImg is for a model trained with the radar rendered on the image.

## 5    CONCLUSION

We have shown that adaptive block-based CS using the prior image and radar data aided in the sharper reconstruction of radar data. In algorithm 1, we used the prior image data to distribute a higher sampling rate on important blocks. The objects that were either missed by the image or the object detection network was effectively captured by the previous radar frame and were reconstructed with a higher sampling rate in algorithm 2. However, the algorithm 2 is provided as a mitigation measure to avoid the scenario of objects being missed by the object detection network or are present in the blindspot of the camera. As the performance of object detection approaches 100% accuracy in the future and with additional camera information, this method can be implemented efficiently with algorithm 1. Also, our algorithm did not exhibit propagation of error in radar reconstruction. That is, although we used the CS reconstructed radar data as prior information in addition to object detection results for algorithm 2, that did not degrade the reconstruction performance of the future frames. Although radar is robust to adverse weather conditions while camera may not be, even during normal weather conditions, it is best to have multiple sensors as they improve performance. In such a situation, we are applying compressed sensing to reduce the acquisition load on the edge device. We believe that during adverse weather conditions, the CS-based radar acquisition could be turned off to acquire the full radar data and gain the relevant information. The numerous other weather sensors and weather prediction tools could aid in this process. However, for a region that is predominantly sunny and does not experience adverse weather conditions, it is resources over-utilization to acquire the radar data at full sampling rates. Our end-to-end transformer based model trained on image and radar has better object detection performance than Faster R-CNN and transformer-based model trained on just images, validating the necessity for radar in addition to images. Similar to image data aiding in sampling radar data efficiently, this method *could* be extended to other modalities. Where, if an object's location is predicted by radar, it could help in sampling LiDAR data efficiently.

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
