# OpenReview forum: "Adaptive Automotive Radar data Acquisition"
_ICLR.cc/2021/Conference — Reject_

### Official Review · AnonReviewer3 · 2020-10-25
**An interesting approach to CS but seems hack-y; need more analysis**

**Rating:** 4
**Confidence:** 4

**Review:**

Summary:
In the paper, the author(s) propose 1) a method to dynamically adjust the sampling rate on radar data using 2D object detections (algo1) and previous image and radar data (algo2); 2) an end-to-end transformer-based 2D object detection model using both radar and image data. The author(s) show experimental results on Oxford Radar RobotCar dataset.
-----------------
Pros:
1. The paper is easy to follow and well written.
2. The author(s) empirically evaluate the proposed DETR model on NuScenes dataset and it beats the faster RCNN baseline.
-----------------
Cons:
1. Magic numbers?: in paragraph 2 of section 4, the author(s) list the way to split the regions and sampling rates for different regions, but does not explain the reason to do so. In my view, this is what the author(s) claim as a novelty “to dynamically allocate sampling rates on the different region”. It is somewhat hack-y to me and hurts the novelty of this paper. Thus I would expect the author(s) can show more solid motivations and reasons for using such a split and allocation method.
2. The evaluation in table 1 looks too subjective to me. If I understand correctly, this evaluates the quality of different reconstruction methods qualitatively. Is this evaluation from one person? And I am not sure if the visual quality of reconstruction has a direct and strong correlation with detection quality. Btw it seems to me it is hard to tell the difference between the reconstruction visualizations in Figure 2.
3. Need more analysis on the latency. In algorithm 1&2, the proposed method uses detection results about 0.2s  before to aid the CS. Assume a car moves 30 mph, in 0.2s it can move 2.68m, which is about 3/4 length of a car. Would this affect the performance?
4. In the evaluation of DETR (table2), do you use the image + CS radar or just image + plain radar? I think it would be good if you can also test the detection performance using image + CS radar.
5. Could you provide the motivation of using the transformer rather than faster RCNN on detection? There are numerous differences between these two, but it is unclear from the paper which one plays the main role in boosting the performance.
-----------------
Questions:
1. Could you explain more about why the block size is 50x100? On page 3, “The radar data is split into 8 equal regions, in azimuth and 37 equal regions in range.” Should the block size be 8 x 37?
Also please address and clarify the cons above
-----------------
Post-rebuttal review:
I carefully read through the rebuttal and other reviews and  I would stick to my current rating.
1. Cons 1: The rebuttal does not provide sufficient explanation on how the magic numbers are chosen (though it is somewhat explained in the rebuttal, it looks more like a design and lacks experiments to back it up: why these numbers but not other numbers?)
2. Cons 2: I would suggest the author(s) further improve the evaluation metrics to make it more objective and convincing.
3. Cons 3: I believe the 2.68m difference in detecting cars should be regarded as a very large error (under IoU 0.5 metric, it would be counted as a misdetection) and should be handled properly.
I think the approach presented in this paper is interesting, and I encourage the author(s) to do more analysis to make it more solid.

---

> ### Author Response · Authors · 2020-11-25
> **Response to AnonReviewer3**
>
> We thank the reviewer for their time and useful comments.
>
> Cons 1: There are three main design choices, first, the radar regions were split into 50x100 blocks because each block covers 45 degrees in azimuth and 4.38m in range. The more granular we get, the better so that it will help in saving on sampling rate by using the increased sampling rate only on important regions. Moreover, with this block size, the CS acquisition can be parallelized for faster processing and reconstruction of the entire radar frame. If we further reduce the size, it could lead to fewer sparse parameters in the block and poor reconstruction using the basis pursuit algorithm in CS. Second, regions split of R1, R2 and R3. The region R1 is the closest to the autonomous vehicle and it was chosen based on the object detection results. The region R2 is also close to the vehicle but, according to the object detection results (algo 1) and object detection results + previous radar (algo 2), it does not have an object. Finally, region R3 is the area far off from the autonomous vehicle. Finally, the last design choice was the sampling rate. We fixed the regions sampling rate for region R3 as 2.5% since it was far away and the sampling rate of region R2 as 5% since it did not have important objects and finally solved for the sampling rate of R3.
>
> Cons 2: Yes, the evaluation is indeed subjective because the Oxford radar dataset is the only publicly available raw radar dataset and object annotation on the radar is not available in that dataset. Therefore, we had to manually evaluate the reconstruction quality of the radar. Moreover, the reconstructed values are grayscale and hence, if the object is not visible in the image, the value of it would be relatively so low that it wouldn’t be detected using CFAR.
> In future work, we are planning to make the evaluation automated by using CFAR or other detection techniques across the entire dataset. Also, in figure 2, if zoomed in, it can be seen that the vehicle to the top-right is reconstructed better using algorithm 1 and algorithm 2. But, the reconstruction quality is poor in baseline reconstruction.
>
> Cons 3: In terms of latency, based on azimuth, we are taking the entire range of 78m in Algorithm 1 and 61.32m in Algorithm 2. Therefore, even if the car had moved 2.68m, it can be captured in this range.
> If the car is far away from the autonomous vehicle i.e., 75m or 58m, that would not be so significant in decision making as it would be captured in the next frame if the vehicle is moving closer to that object or can be ignored if it is moving away.
>
> Cons 4: In the evaluation of DETR, I used only image + plain radar because the CS radar analysis was performed on the Oxford radar dataset (raw radar) and it is significantly different than NuScenes in terms of radar density and so on. Also, DETR required a significant amount of radar data to be trained on with annotations which were not available in the case of raw radar data from the Oxford dataset. In future work, we plan on using semi-supervised techniques to annotate the Oxford dataset and use them for DETR training instead of NuScenes.
>
> Cons 5: The first analysis of Algorithm 1 and Algorithm 2 used only CS techniques for reconstruction and manual annotation. In this part, we used Faster R-CNN for object detection. In the second part, to the best of our knowledge, we were the first ones to show that radar + image for object detection on the transformer-based detection network performed better than Faster R-CNN. The main advantage of transformers is that their architectural complexity is simpler than Faster R-CNN and the need for Non-Maximal suppression is eliminated in DETR. Also, we believe that the attention mechanism in DETR helped in focusing more on the regions with radar points (overlapping on objects in images) and helped in better performance.
>
> Question 1: Yes, the total block size from the original radar is 8x37. But, each block has 50x100 pixel values/radar values and hence, we called the block size to be 50x100. As stated in the reply to cons1, the size 50x100 was ideal with more sparsity to use CS efficiently and at the same time, this granularity would help in parallelizing the CS acquisition and reconstruction.

---

### Official Review · AnonReviewer4 · 2020-10-28
**Recommendation to Reject due to Lack of Algorithmic Novelty**

**Rating:** 3
**Confidence:** 3

**Review:**

##########################################################################
Summary:

The paper develops a method to select a radar return region to be sampled at a higher rate based on a previous camera image and radar recording. Furthermore, the paper validates that an end2end transformer model trained on both camera and radar data outperforms an end2end transformer model only trained on camera data and hence, supports the argumentation to add a radar sensor to an automated vehicle.

##########################################################################
Reasons:

Overall, I vote for rejecting the paper. While it is generally a great idea to guide the selection of radar regions to be sampled at a higher rate the paper is very application-focused and lacks novelty in its method. The result that camera and radar data combined will outperform camera data only is expected. Using detections in images to guide the radar reverses its advantage to work well in adverse weather conditions compared to the camera.

##########################################################################
Pros:

* Interesting and relevant topic
* Training results support claims

##########################################################################
Cons:

* Lack of algorithmic novelty
* Use of Faster R-CNN (slow)
* Using camera to select most important radar regions contradicts the stated advantage of radars to perform better in adverse weather conditions

---

> ### Author Response · Authors · 2020-11-25
> **Response to AnonReviewer4**
>
> We thank the reviewer for their critical and insightful comments.
> As stated correctly, the algorithm novelty is predominantly in guiding radar acquisition using object detection results from images. However, it is vital to study this feedback process of using previous image data for efficient radar data acquisition. There are numerous studies on CS-based radar acquisition. But, we are the first ones to use Object detection results to guide efficient radar data acquisition and show that with as little as 10% sampling rate, we can recover the objects near the autonomous vehicle.
> Also, in DETR we showed that using Radar on Images, and an attention network-based object detection network has improved detection rate.
>
> Apart from the stated advantage of the radar in adverse weather conditions, during normal times, it is best to have multiple sensors as they improve performance. In such a situation, we are applying compressed sensing to reduce the acquisition load on the edge device. We believe during adverse weather conditions, the CS-based radar acquisition could be turned off to acquire the full radar data and gain the relevant information. The numerous other weather sensors and weather prediction tools could aid in this process. However, for a region that is predominantly sunny and does not experience adverse weather conditions, it is resources over-utilization to acquire at full sampling rates.
>
> In an NVIDIA GPU, faster R-CNN takes about 0.12s per image to process. Based on the average speed of 40 miles per hour in an urban environment, the vehicle or object could have moved about 2.15m which would be covered by focusing on the first 78m (Algorithm 1) and 61m (Algorithm 2). However, we agree that we should deploy other faster object detection networks that consume less than 0.12s per image.

---

### Official Review · AnonReviewer1 · 2020-10-30
**Review for the paper: Adaptive Automotive Radar data Acquisition**

**Rating:** 4
**Confidence:** 4

**Review:**

The authors propose an algorithm to select radar return regions that potentially contain objects inside.

Strengths:
+ It seems to be an interesting topic that uses radar data together with RGB images from the camera.

Weaknesses:
- The problem to solve in this paper is not reasonably stated. The authors claim that the purpose of detecting the "important regions" in the radar domain is to implement compressed sensing during radar data acquisition. However, since object detection on radar data is already proposed and studied by some researchers (e.g., [1][2][3]), the reason why this problem needs to be proposed separately is not clear to me.
- The innovation is limited for both Algorithm 1 and 2. It seems that algorithms that trying to provide some bounding boxes on radar from Faster R-CNN and CFAR detections.
- The experiments are not adequate to illustrate the performance. It's not clear that the dataset used in this paper. The radar points in the nuScenes dataset are significantly different from the Oxford radar dataset, especially on the density of the radar points due to different kinds of radar sensors. Besides, the results in Table 1 seems very simple by selecting some special cases from the testing set.

Overall, I think the paper is not good enough to be accepted by ICLR.


[1] Major, Bence, et al. "Vehicle Detection With Automotive Radar Using Deep Learning on Range-Azimuth-Doppler Tensors." Proceedings of the IEEE International Conference on Computer Vision Workshops. 2019.
[2] Nobis, Felix, et al. "A Deep Learning-based Radar and Camera Sensor Fusion Architecture for Object Detection." 2019 Sensor Data Fusion: Trends, Solutions, Applications (SDF). IEEE, 2019.
[3] Wang, Yizhou, et al. "RODNet: Object Detection under Severe Conditions Using Vision-Radio Cross-Modal Supervision." arXiv preprint arXiv:2003.01816 (2020).

---

> ### Author Response · Authors · 2020-11-25
> **Response to AnonReviewer1**
>
> We thank the reviewer for their time and for stating the strengths and weaknesses clearly.
>
> Weakness 1: Yes, radar object detection is an active field with numerous studies. However, In this paper, we have focused on radar data acquisition which is an expensive process. For example, in the Oxford radar dataset, each radar frame has about 1.48 million data points and they are acquired at a 4Hz rate. At this rate, the radar acquisition consumes significant power and memory on the edge acquisition device. We are proposing compressed sensing in order the reduce the number of datapoint being acquired and hence save on power and time during acquisition. Therefore, this study has been primarily proposed as an efficient radar acquisition algorithm using previous object detection results from images.
>
> Weakness 2: We agree that the major innovation in the paper is in algorithm 1 and algorithm 2. However, as stated in the paper to the best of our knowledge we are the first ones to implement radar + image object recognition using transformer-based Object detection network and show that the performance is better than only image-based techniques.
>
> Weakness 3: Algorithm 1 and algorithm 2 were evaluated on the Oxford Radar dataset. The object detection using transformer networks was independently trained and evaluated on the NuScenes dataset. Due to the lack of radar object annotation from the Oxford Radar dataset, we performed the manual evaluation and hence we had to limit our analysis to 3 random scenes from the Oxford Radar dataset. Moreover, for algorithm 1 and algorithm 2, there is no training set or test set. Mainly because, we compress the radar data, reconstruct and evaluate if the object is present or not. So, ideally, all the three scenes we chose were “testing” set only.

---

### Official Review · AnonReviewer2 · 2020-10-31
**A lot of ad-hoc choices that are not well-reasoned**

**Rating:** 4
**Confidence:** 4

**Review:**

The paper proposes a sensor fusion approach combining radar and camera to improve the detection of object in an automotive sensing scenario.

The paper proposes two different algorithms to solve this problem. It is not clear when one should be preferred over the other. It seems that the second algorithm is an improvement over the first. The first one uses only the camera data and also has  a partial blind spot. The second includes the radar data and improves on the first one. It seems to me that the description of the two algorithms lacks focus. Since the second algorithm is the one that the authors are trying to promote, why not focus on that one?

There are also a lot of ad hoc choices that are not justified. For example, they pick 18 block which determines the range of the radar. Why not 20? Why not 16? Similarly, there is a splitting of the scene into regions, which is also different in algorithm 1 and algorithm 2. Why is that? How was the number and extent of regions chosen? There is no discussion.

Overall the paper seems like a collection of ad-hoc engineering choices. There is no discussion or reasoning in these choices. While the paper might have some nuggets of interesting ideas, these are buried in the detail of all these design choices of the authors. I would recommend a complete restructuring of the paper to clearly expose the goals and contributions of the authors, rather than just list a set of unmotivated design choices.

Also, given the heuristic nature of such designs, I would recommend the papers provide much more extensive experimental evaluation, especially in comparing their approach with other approaches, rather than just one.

---

> ### Author Response · Authors · 2020-11-25
> **Design choices explained**
>
> We thank the reviewer for understanding the paper and for their valuable feedback.
>
> We develop a novel algorithm motivated by the hypothesis that with a limited sampling budget, allocating more sampling budget to areas with an object as opposed to a uniform sampling budget improves radar data acquisition quality.
> The primary motivation of the paper is to use image data to guide radar data acquisition by focusing on the blocks where the object has appeared. The major novelty factor is the usage of the object detection network’s output as prior information for adaptive compressed sensing of radar data.
>
> It is a valid point that we could have focused only on the second algorithm. We intended on presenting the second algorithm as an improvement to the first algorithm. During the first algorithm design, as you correctly mentioned, we encountered blindspots in which objects could be present. To tackle such a situation, we improved on the first algorithm to include previous radar points as well and perform compressed sensing. This is the logical flow with which we came up with both the algorithms. In the corrections, we will reduce the emphasis on the first algorithm and focus more on the second algorithm.
>
> In general, the first design choice of choosing 18 blocks was to focus on the minimum required range where half of the available information (37 blocks - 163m) corresponding to 78.84m. Since the depth information is not available from the image, if we choose the entire 163m for a particular azimuth block, we will waste the sampling budget on distant objects. Therefore, we fathomed that 78m in the case of Algorithm 1 and 61m in the case of Algorithm 2 is a safe choice as the distance to be covered based on the explanation given in the paper. If we increase this, we may potentially waste budget allocation on distant objects and if we chose fewer blocks, it could hinder safely determining objects near the autonomous vehicle.
>
> The only difference between scene split in algorithm 1 vs. algorithm 2 was to focus on the first 14 blocks (61.32m) rather than the first 18 blocks (78.84m). From images, we could find only the azimuth of the object and not the depth. Hence, for a particular azimuth, we are forced to take the first 61m or 78m for which we’ve justified. Therefore, the budget we save by not considering the blocks from 14th to 18th in Algorithm 2 was used to redistribute to other important regions as indicated by the radar pointcloud.
>
> As suggested by the reviewer, we have performed an additional baseline analysis of adaptive CS using only radar pointcloud as prior information and we have included that in the paper.
>
> We thank the reviewer for the suggestion of restructuring the paper to highlight our contributions and we will do so.

---

### Decision · Program_Chairs · 2021-01-07
**Final Decision**

**Decision:**

Reject

**Comment:**

The main idea of the paper is to  use image data to guide radar data acquisition by focusing on the blocks where the object has appeared. Four reviewers have relatively consistent rating: 3 of them rated “Ok but not good enough - rejection”, while 1 rated “clear rejection”. The main concerns include ad-hoc choices of algorithm design, lack of algorithm novelty, not adequate experiments in illustrating the performance, etc. During the rebuttal, the authors made efforts to response to all reviewers’ comments. However, the major concerns remain, and the rating were not changed. While the motivation is clear and the work has merits, the ACs agree with the reviewers’ concerns and this paper can not be accepted at its current state.